# Retrieval-Aware Distillation for Transformer-SSM Hybrids

Aviv Bick [1]   Eric P. Xing [1 2]   Albert Gu [1 3]

## Abstract

State-space models (SSMs) offer efficient sequence modeling but lag behind Transformers on benchmarks that require in-context retrieval. Prior work links this gap to a small set of attention heads, termed Gather-and-Aggregate (G&A), which SSMs struggle to reproduce. We propose *retrieval-aware distillation*, which converts a pretrained Transformer into a hybrid student by preserving only these retrieval-critical heads and distilling the rest into recurrent heads. We identify the essential heads via ablation on a synthetic retrieval task, producing a hybrid with sparse, non-uniform attention placement. We show that preserving **just 2% of attention heads recovers over 95% of teacher performance on retrieval-heavy tasks** (10 heads in a 1B model), requiring far fewer heads than hybrids that retain at least 25%. We further find that large recurrent states often compensate for missing retrieval: once retrieval is handled by these heads, the SSM backbone can be simplified with limited loss, even with an $8\times$ reduction in state dimension. By reducing both the attention cache and the SSM state, the resulting hybrid is $5$–$6\times$ more memory-efficient than comparable hybrids, closing the Transformer–SSM gap at a fraction of the memory cost.

## 1. Introduction

State-space models (SSMs) show strong language modeling capabilities with high efficiency, but their constant-memory design causes them to underperform Transformers on benchmarks that require referencing earlier context—a function known as in-context retrieval (Arora et al., 2023; Bick et al., 2025b). To bridge this gap, recent hybrid architectures combine SSM backbones with a small number of attention heads that cache the full sequence history. Recently, Bick et al. (2025b) showed that retrieval capabilities in such hybrid models naturally concentrate in attention heads termed Gather-and-Aggregate (G&A), undertaking the burden of retrieval from SSMs and allowing them to focus on general language modeling. At the same time, their findings suggest that only a few of these heads are needed, implying that many existing hybrids retain substantially more attention than functionally necessary and therefore incur avoidable memory overhead.

Motivated by this observation, we introduce *retrieval-aware distillation*, a strategy that transforms pretrained Transformers into hybrids by explicitly preserving the retrieval components that SSMs struggle to learn, yielding $5$–$6\times$ memory savings over comparable hybrids. As illustrated in Figure 1, we first score attention heads by ablation on a synthetic KV-retrieval task and keep only the high-scoring ones. We then replace the remaining attention heads with SSM-based recurrent heads and perform standard distillation into the resulting hybrid student. Unlike prior hybrid distillation methods (Wang et al., 2025b; Glorioso et al., 2024; Dong et al., 2024) that use pre-set attention layouts (e.g., fixed ratios or alternating layers), retrieval-aware distillation selects and preserves specific individual attention heads based on empirical retrieval functionality, thereby reducing overhead without sacrificing performance.

We apply retrieval-aware distillation to Llama-3.2-1B and Qwen-2.5-1.5B using the MOHAWK distillation framework (Bick et al., 2024). Across both models, *retaining just 10 attention heads (2% of 512) recovers over 95% of the full Transformer's performance*, with little improvement beyond that. By comparison, prior distillation baselines (Wang et al., 2025b; Bick et al., 2024) require *at least* 25% of heads to reach similar performance, reducing the attention-head budget by over $10\times$.

We confirm that the gains come from restored retrieval by analyzing hybrids with different numbers of retained G&A heads. Retaining only the top 10 retrieval heads markedly improves retrieval-intensive benchmarks over a distilled SSM-only student (no retained attention heads), e.g., *SWDE: 27.7%→71.1%, KV-Retrieval: 13.2%→99%*, and yields substantial perplexity gains—enough to close most of the perplexity gap on retrieval-intensive tokens (Arora et al., 2023). Attention-map visualizations support this interpre-

[1]Carnegie Mellon University [2]MBZUAI [3]Cartesia AI. Correspondence to: Aviv Bick <abick@cs.cmu.edu>.

*Proceedings of the 43rd International Conference on Machine Learning*, Seoul, South Korea. PMLR 306, 2026. Copyright 2026 by the author(s).

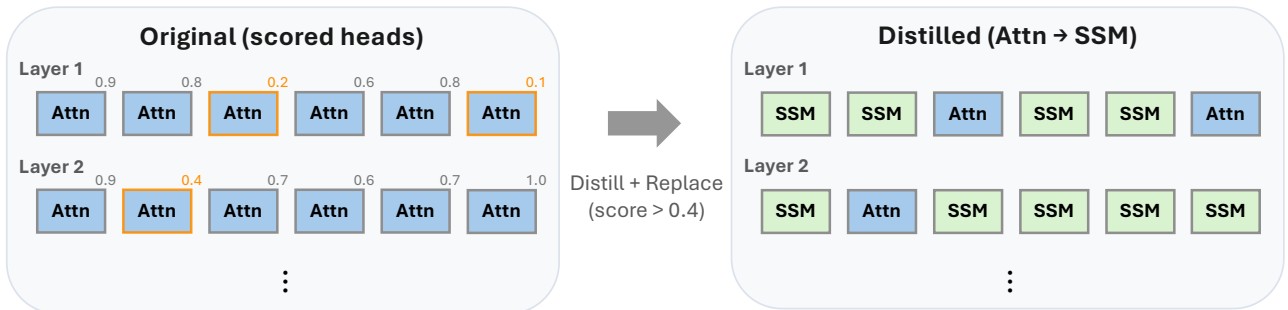

*Figure 1.* **Retrieval-Aware Attention Placement During Distillation.** We add a retrieval-guided step before standard distillation: (1) ablate each attention head in the pretrained Transformer and measure the accuracy drop on a synthetic KV-retrieval probe to obtain a retrieval-importance score; (2) retain only heads above a threshold and replace the rest with recurrent heads; (3) distill into a hybrid student. We use KV-retrieval only for head ranking; it identifies the same G&A heads that drive retrieval-intensive performance (Bick et al., 2025b). Unlike heuristic hybrids, this yields non-uniform attention placement that preserves performance with far fewer heads and reduces attention's KV-cache bandwidth cost (Dao et al., 2022).

tation, showing that G&A operations concentrate in the preserved heads, consistent with the specialization observed in full-layer hybrids (Bick et al., 2025b).

Moreover, we show that retrieval-aware hybrids allow a drastically simpler SSM backbone. In particular, the SSM state dimension can be reduced by $8\times$ (e.g., from 64 to 8) with limited performance degradation. This suggests that large state dimensions in pure SSMs often compensate for missing retrieval capability; once attention assumes this role, smaller states suffice.

Overall, by reducing both the attention cache and the SSM state, retrieval-aware hybrids are $5\times$ more memory-efficient on short sequences (128 tokens) and up to $6\times$ on long sequences (4K tokens) relative to other hybrids, with corresponding gains in prefill and decoding throughput.

## 2. Related Work

**Hybrid Language Models.** Transformer-based language models (Vaswani et al., 2017; Touvron et al., 2023) remain the foundation of modern language modeling due to their scalability and empirical performance; however, their quadratic computational complexity makes them prohibitively expensive for long sequences. Recurrent models, such as state-space models (SSMs) (Gu & Dao, 2023; Dao & Gu, 2024), offer an efficient alternative with linear time complexity, yet they typically incur a performance gap compared to full Transformers. To bridge the divide between SSM efficiency and Transformer performance, several hybrid architectures have been proposed (Lieber et al., 2024; Dong et al., 2024; Glorioso et al., 2024). For instance, Glorioso et al. (2024) employs shared attention layers across blocks to reduce parameter counts, whereas Dong et al. (2024) combines quadratic and sliding-window attention with SSMs in a fixed 1:5 ratio. While these designs miti-

gate the memory-bandwidth bottlenecks of full Transformers, they typically place attention using fixed, layer-level patterns—e.g., a predetermined attention–SSM mixing ratio or a fixed interleaving of attention and SSM blocks. As a result, they may allocate attention capacity inefficiently, retaining more attention heads (and incurring higher costs) than necessary to preserve performance.

**The Retrieval Gap in SSMs.** A growing body of work has identified in-context retrieval as the primary differentiator between Transformer and SSM performance (Arora et al., 2023; Jelassi et al., 2024; Park et al., 2024; Bick et al., 2025b). Wen et al. (2024) highlight SSM weaknesses in associative recall, while Jelassi et al. (2024) provide theoretical evidence that SSMs struggle with precise copying operations. Crucially, Bick et al. (2025b) localized this limitation, attributing the performance gap to a small subset of "Gather-and-Aggregate" attention heads. This insight suggests that the retrieval gap is not a holistic failure of the architecture, but a specific functional deficit. Our work operationalizes this finding by preserving only those specific heads essential for full-context retrieval, replacing the remainder with SSM heads for efficient modeling.

**Distillation for Hybrid models.** Distillation has become a key technique for initializing hybrid models from pretrained Transformers (Paliotta et al., 2025; Wang et al., 2025a; Bick et al., 2025a). Wang et al. (2025b) utilize the State-Space Duality (SSD) framework (Dao & Gu, 2024) to reuse linear projection weights; concurrently, Bick et al. (2024) propose a multi-stage pipeline involving matrix orientation and hidden-state alignment. However, existing literature largely focuses on the *method* of weight transfer (how to distill) rather than the *architecture design* (what to distill into). These approaches typically assume a predetermined student architecture with fixed attention patterns. There remains lim-

ited exploration into using the teacher's internal structure to inform the student's design, specifically by identifying and preserving functional components during the distillation process.

## 3. Background

### 3.1. Notation and Token Mixing

Let $X \in \mathbb{R}^{T \times d}$ denote a length-$T$ sequence of token representations, with $x_t \in \mathbb{R}^d$ the representation at position $t$. We view both attention and SSM blocks as *token-mixing* operators (mix across positions) followed by standard pointwise transformations (e.g., MLPs and residual connections).

**Attention (global mixing).** For a given head, the block forms queries, keys, and values via learned linear projections of $X$ (we omit the projection matrices for brevity), and mixes tokens by

$$\text{Attn}(X) = \text{softmax}\left(\frac{QK^\top}{\sqrt{d_k}}\right)V,$$

where $Q, K \in \mathbb{R}^{T \times d_k}$ and $V \in \mathbb{R}^{T \times d_v}$ are the projected queries, keys, and values. This explicitly enables long-range retrieval by allowing each position to attend to earlier tokens.

**State-space models (recurrent mixing).** An SSM block maintains a latent state $h_t$ that summarizes the past and is updated sequentially:

$$h_t = A_t h_{t-1} + B_t x_t, \qquad y_t = C_t h_t + D_t x_t,$$

where $A_t, B_t, C_t, D_t$ are learned parameters (often projections of $x_t$ in practice) and $y_t \in \mathbb{R}^d$ is the mixed output at position $t$. Unlike attention, this recurrence uses constant memory in $T$, but represents history through a compressed state, which can limit exact retrieval of specific past items (Arora et al., 2023; Bick et al., 2025b).

**Matrix-based token mixers.** Both mechanisms can be written as applying a (possibly input-dependent) mixing matrix over the sequence dimension. Concretely, a token-mixing block produces

$$Y = M(X)X,$$

where $M(X) \in \mathbb{R}^{T \times T}$ specifies how each output position combines information from all input positions. For attention, $M(X) = \text{softmax}\left(\frac{QK^\top}{\sqrt{d_k}}\right)$ is computed from the input and is generally dense. For SSMs, the recurrence induces a structured mixing matrix $M$ (typically independent of $X$ for linear time-invariant SSMs) that is efficiently applied without explicitly forming the full $T \times T$ matrix, yielding constant-memory computation in $T$.

This shared matrix viewpoint enables a direct comparison between attention heads and SSM blocks as token mixers, and forms the basis of the Gather-and-Aggregate analysis, which studies how specific heads implement retrieval-relevant mixing patterns (Bick et al., 2025b).

### 3.2. Gather-and-Aggregate Heads

Bick et al. (2025b) identified a small subset of mixing heads responsible for in-context retrieval in both Transformer and SSM-based language models. These heads implement a shared computational pattern termed the *Gather-and-Aggregate (G&A)* mechanism. To illustrate this, consider a "Dictionary Learning" task where the model must retrieve a value (e.g., 50) associated with a specific key (e.g., scallops) from a context of key-value pairs. The G&A mechanism coordinates two distinct roles to solve this:

- **Gather Heads:** These heads compress local information into "transport" tokens. For example, in the segment "scallops:50\n", a Gather head moves the semantic information from scallops and 50 into the newline token (\n), effectively creating a summary vector at that position.

- **Aggregate Heads:** These heads perform the global retrieval. To predict the final answer, the Aggregate head attends primarily to the "transport" tokens (the \n positions) across the sequence, identifying the one matching the query and extracting its stored value.

Bick et al. (2025b) demonstrated that while Transformers perform this explicitly via attention, SSMs approximate it implicitly. Crucially, they showed that this mechanism can be isolated: the specific heads responsible for G&A can be detected by measuring the performance drop when individual heads are ablated—a property we leverage directly in our architecture design (see Section 4).

### 3.3. MOHAWK Distillation

MOHAWK (Bick et al., 2024) distills a pretrained Transformer into an SSM by treating both as *sequence mixing operators*, which enables direct teacher–student matching at multiple granularities. The pipeline optimizes the student parameters ($\phi$) across three stages:

**Matrix Orientation.** For each layer $\ell$, MOHAWK matches the teacher attention mixer and the student SSM mixer by minimizing the Frobenius distance between their induced token-mixing operators,

$$\min_{\phi_\ell} \mathbb{E}_u\left[ \|M_T^{(\ell)}(u) - M_S^{(\ell)}(u; \phi_\ell)\|_F \right],$$

where $u$ is taken from the teacher's layer-$(\ell-1)$ output so both mixers are evaluated on matched inputs. Here $M_T(u) \in \mathbb{R}^{L \times L}$ denotes the attention weight matrix, while $M_S(u)$ denotes the corresponding SSM mixing operator (typically implicit, and materialized only when needed).

**Hidden-State Alignment.** After orienting the mixers, MO-HAWK aligns the *block outputs* (e.g., an attention block vs. an SSM/mixer block) by minimizing an $L_2$ distance,

$$\min_{\phi_\ell} \mathbb{E}_u \left[ \|\text{Block}_T^{(\ell)}(u) - \text{Block}_S^{(\ell)}(u; \phi_\ell)\|_2^2 \right],$$

again using teacher-derived inputs $u$ so layers can be trained independently (and in parallel).

**Weight Transfer & Knowledge Distillation.** Finally, MO-HAWK transfers the remaining non-mixer weights (e.g., MLP/norm/embeddings) from the teacher and finetunes the full student end-to-end with a logit distillation objective,

$$\min_{\phi} \mathscr{L}_{\text{CE}}\big(\text{Model}_T(x), \text{Model}_S(x; \phi)\big).$$

Bick et al. (2024) also introduce DISCRETEMAMBA2, a discretized Mamba-2 variant used to improve compatibility between attention-style and SSM-style mixing during these alignment stages; we use it as the basis for our SSM replacements.

### 3.4. Evaluation Benchmarks

Following Bick et al. (2025b), we assess how in-context retrieval affects downstream performance by dividing benchmarks into two categories. The first group, *Knowledge-Focused Tasks*, includes PIQA (Bisk et al., 2019), Winogrande (Sakaguchi et al., 2019), OpenBookQA (Mihaylov et al., 2018), HellaSwag (Zellers et al., 2019), and both ARC-Challenge and ARC-Easy (Clark et al., 2018). These tasks primarily test factual knowledge and commonsense reasoning, with minimal dependence on retrieving information from the surrounding context.

In contrast, the second group comprises *Retrieval-Heavy Tasks*, which place a stronger emphasis on multi-hop reasoning, mathematical deduction, or sequence-level recall. This set includes Lambada (Paperno et al., 2016), MMLU (Hendrycks et al., 2021), GSM8K (Cobbe et al., 2021), SWDE (Arora et al., 2024), and synthetic KV-Retrieval (Bick et al., 2025b).

To summarize performance within each category, we additionally report *coverage* with respect to the teacher model. For each benchmark group, coverage is defined as

$$\text{Coverage}(\%) = 100 \cdot \frac{\frac{1}{|G|} \sum_{t \in G} s_{\text{model}}(t)}{\frac{1}{|G|} \sum_{t \in G} s_{\text{teacher}}(t)},$$

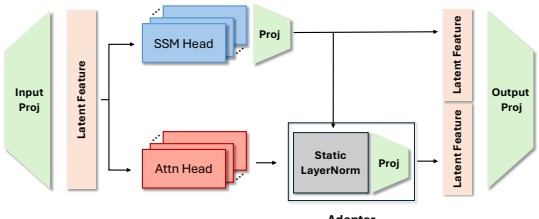

*Figure 2.* **Transformer-SSM Integration.** We selectively replace attention heads with SSMs. To ensure distributional alignment, we normalize the remaining **concatenated head states** to match the mean and variance of the SSM output **before the final projection** (*Static LayerNorm* in the figure). This parameter-free step stabilizes hybrid integration. Code in Section A.2.

where $G$ is either the knowledge-focused or retrieval-heavy task set, and $s_{\text{model}}(t)$ and $s_{\text{teacher}}(t)$ denote accuracy on task $t$. This metric measures what fraction of the teacher's average performance is retained within each benchmark group. Note that Coverage can exceed 100% when the student slightly outperforms the teacher on the group mean.

## 4. Methods

Our contribution is a novel architectural selection method that constructs efficient hybrid models by strictly preserving the attention heads responsible for the G&A mechanism. Unlike prior hybrids that retain attention layers at fixed intervals (e.g., every $n$-th layer), our approach is targeted. Our method consists of two main steps: identifying retrieval-critical heads and constructing a hybrid architecture around them. Once constructed, we train the model using a modified version of the MOHAWK framework.

### 4.1. Step 1: Identifying G&A Heads via Ablation

The first step is to isolate the specific attention heads in the teacher model that perform Gather-and-Aggregate operations. We assume that heads critical for G&A will exhibit high sensitivity during retrieval tasks. Following Bick et al. (2025b), we employ a synthetic Key-Value (KV) retrieval task on the pretrained Transformer teacher. We systematically ablate each attention head in the teacher model individually (masking its output to zero) and measure the resulting drop in accuracy on the synthetic task. We rank all heads by their "retrieval importance score," defined as the magnitude of the performance drop when ablated. This provides a sorted list of heads, from most to least critical for in-context retrieval.

### 4.2. Step 2: Hybrid Model Construction

Given the per-head ranking from Step 1, we construct a student that *retains* only the top-$k$ attention heads, where $k$ is a user-chosen budget controlling the attention footprint;

*Table 1.* **Hybrid models recover teacher performance with few retrieval-critical heads.** We report accuracy for HYBRID-LLAMA-1B and HYBRID-QWEN-2.5-1.5B as we vary the number of retained attention heads; teacher models and full transformer-to-transformer distillation are shown for reference. For each $k$, the hybrid keeps the top-$k$ heads ranked by KV-Retrieval ablation importance, so smaller $k$ retains only the most retrieval-critical G&A heads. Benchmarks are grouped into knowledge-focused and retrieval-heavy tasks (Section 3.4). Coverage (COV) is computed per group as the ratio of the model and teacher group-mean scores. Shaded rows highlight the rapid recovery from $k=0$ to $k=10$, where coverage above 95% is treated as sufficient.

| MODEL | #ATT HEADS | KNOWLEDGE-FOCUSED | | | | | | | RETRIEVAL-HEAVY | | | | | |
|---|---|---|---|---|---|---|---|---|---|---|---|---|---|---|
| | | ARC-C | ARC-E | PIQA | WG | HS | OBQA | COV | LMB | MMLU | GSM8K | SWDE | KV-Ret | COV |
| **Hybrid-Llama-1B** | 0 | 38.0 | 69.3 | 74.2 | 61.7 | 61.0 | 36.6 | **101.4** | 50.7 | 39.2 | 25.1 | 27.7 | 13.2 | 49.2 |
| | 5 | 36.8 | 69.4 | 73.7 | 60.8 | 58.8 | 36.8 | **100.1** | 53.6 | 40.3 | 30.0 | 66.0 | 90.0 | 88.3 |
| | 10 | 37.6 | 69.0 | 74.6 | 60.5 | 62.0 | 36.8 | **101.3** | 54.2 | 42.1 | 34.4 | 71.1 | 99.0 | **95.0** |
| | 20 | 38.2 | 69.3 | 74.5 | 62.9 | 61.1 | 36.5 | **101.9** | 55.0 | 43.0 | 34.0 | 72.5 | 99.3 | **95.8** |
| | 30 | 39.3 | 69.3 | 75.0 | 61.5 | 62.2 | 38.4 | **102.9** | 54.0 | 43.4 | 33.1 | 70.4 | 98.0 | **94.3** |
| | 40 | 37.5 | 68.9 | 73.7 | 61.8 | 59.2 | 37.6 | **100.8** | 54.0 | 44.0 | 34.0 | 71.1 | 99.4 | **95.4** |
| | 512 | 37.8 | 70.1 | 74.4 | 61.7 | 62.7 | 39.4 | **103.0** | 56.0 | 45.0 | 35.8 | 75.3 | 99.4 | **98.2** |
| LLAMA-3.2-1B | 512 | 38.1 | 68.5 | 74.4 | 59.7 | 60.8 | 34.6 | **100.0** | 60.1 | 46.0 | 33.1 | 78.6 | 99.3 | **100.0** |
| **Hybrid-Qwen-2.5-1.5B** | 0 | 46.5 | 76.3 | 75.5 | 64.0 | 65.1 | 40.0 | **99.1** | 53.6 | 51.9 | 26.8 | 32.1 | 16.0 | 52.1 |
| | 5 | 47.1 | 76.0 | 75.8 | 62.9 | 65.7 | 40.2 | **99.2** | 56.6 | 53.2 | 35.3 | 78.0 | 90.0 | 90.4 |
| | 10 | 47.1 | 75.8 | 75.6 | 63.0 | 65.3 | 40.4 | **99.0** | 57.1 | 55.4 | 40.5 | 81.0 | 100.0 | **96.4** |
| | 20 | 45.0 | 76.3 | 76.0 | 62.4 | 65.9 | 41.2 | **98.9** | 57.5 | 56.3 | 40.5 | 81.3 | 100.0 | **96.9** |
| | 30 | 45.9 | 76.4 | 76.1 | 63.4 | 66.1 | 41.2 | **99.5** | 58.0 | 57.0 | 43.1 | 81.6 | 100.0 | **98.1** |
| | 336 | 48.2 | 77.6 | 75.9 | 63.6 | 67.5 | 40.0 | **100.5** | 60.1 | 59.8 | 42.6 | 81.4 | 100.0 | **99.3** |
| QWEN-2.5-1.5B | 336 | 46.4 | 76.6 | 75.9 | 63.3 | 68.2 | 40.4 | **100.0** | 60.0 | 60.1 | 43.8 | 82.5 | 100.0 | **100.0** |

in practice, we find that retaining a small number of heads (often 10–20) captures most of the gains, while larger $k$ mainly increases memory overhead.

The student preserves the teacher's depth and MLPs, but modifies the mixing layers as follows:

- **Retained heads:** The top-$k$ attention heads are kept unchanged (including their $Q, K, V$ and output projections).

- **Replaced heads:** All remaining heads are replaced by a `DiscreteMamba2` component (Bick et al., 2024) with a state dimension matching the original head dimension.

This yields heterogeneous layers: some layers become pure SSM (if none of their heads fall in the top-$k$), while others become hybrids that run attention and SSM in parallel (see Figure 1). For example, with $k=20$, Llama-3.2-1B-Instruct retains three heads in each of `L{0,8,10}`, two heads in each of `L{4,5,7}`, and one head in each of `L{1,3,6,9,11}` (see Table 5).

**Feature Alignment.** We combine the retained attention heads and the new SSM components within a standard multi-head layout. Given the block input, we split its feature dimension into head-sized chunks (one chunk per head). Chunks assigned to retained heads are processed by their original attention operators, while all other chunks are processed by `DiscreteMamba2` modules with the same head dimension. We then concatenate the per-head outputs and apply the usual output projection, producing a single mixing output with the same interface as the teacher.

*Table 2.* **Memory Usage Across Sequence Lengths.** Inference memory (MB) per sequence for hybrids matching the teacher LLAMA-3.2-1B. RETRIEVAL-AWARE minimizes footprint through two mechanisms: (1) a reduced SSM state dimension ($d=8$) lowers constant overhead, and (2) retaining only 2% attention heads reduces the KV cache. This yields substantial savings relative to layer-wise baselines (25% and 50%) in Table 3. Full analysis in Section A.4.

| Model | $L=128$ | $L=2048$ | $L=4096$ |
|---|---|---|---|
| HYBRID-LLAMBA | 0.8 MB (×1.0) | 5.7 MB (×1.0) | 11.0 MB (×1.0) |
| HYBRID-MOHAWK | 4.1 MB (×5.1) | 19.5 MB (×3.4) | 35.8 MB (×3.3) |
| MAMBA-IN-LLAMA | 4.2 MB (×5.1) | 35.7 MB (×6.2) | 69.2 MB (×6.3) |
| LLAMA-3.2-1B | 4.2 MB (×5.1) | 67.1 MB (×11.7) | 134.2 MB (×12.2) |

To reduce distributional mismatch between the attention and SSM outputs before concatenation, we apply a parameter-free LayerNorm to the attention outputs, rescaling them to match the mean and variance of the SSM outputs. This stabilizes their combination in the residual stream without introducing additional learned parameters (see Figure 2 and Section A.2).

## 5. Empirical Validation

We now provide empirical evidence that retaining Gather-and-Aggregate (G&A) heads during distillation from a pre-trained Transformer to a Transformer-SSM hybrid allows the student model to close the performance gap between the two architectures. These results demonstrate that performance gains depend not on *how many* attention heads are used, but on *which* heads are retained and *where* they are

*Table 3.* **Efficiency of Retrieval-Aware vs. Heuristic Placement.** Results on LLAMA-3.2-1B comparing retrieval-aware head placement to **Fixed interleaving** (Bick et al., 2024) and **Annealed interleaving** (Wang et al., 2025b) heuristics. COV reports the ratio between the model and teacher group-mean accuracy (defined in Section 3.4). Coverage depends on whether the heuristic includes retrieval-critical layers; e.g., in FIXED 25%, (4, 9, 14) performs best, matching high-impact retrieval heads in layers 4 and 9 (see *L4H*4 and *L9H*25 in Table 5). Retrieval-aware placement outperforms many variants that use far more attention heads; even when heuristics exceed 95% COV, they require 10×–13× more heads and can select unfavorable layers, leading to suboptimal outcomes.

| METHOD | FULL ATT LAYERS | #ATT HEADS | KNOWLEDGE-FOCUSED | | | | | | | RETRIEVAL-HEAVY | | | | | |
|---|---|---|---|---|---|---|---|---|---|---|---|---|---|---|---|
| | | | ARC-C | ARC-E | PIQA | WG | HS | OBQA | COV | LMB | MMLU | GSM8K | SWDE | KV-Ret | COV |
| LLAMA-3.2-1B | 0-15 | 512 | 38.1 | 68.5 | 74.4 | 59.7 | 60.8 | 34.6 | **100.0** | 60.1 | 46.0 | 33.1 | 78.6 | 99.3 | **100.0** |
| FIXED 25% | 0, 5, 10, 15 | 128 | 37.3 | 69.3 | 74.3 | 61.9 | 59.5 | 36.0 | **100.7** | 56.4 | 44.2 | 35.4 | 75.2 | 99.0 | **97.8** |
| | 0, 4, 8, 12 | 128 | 37.1 | 68.6 | 74.3 | 62.5 | 59.6 | 38.5 | **101.3** | 55.8 | 41.0 | 36.3 | 73.0 | 99.2 | **96.3** |
| | 1, 5, 9, 13 | 128 | 37.5 | 69.2 | 73.6 | 60.4 | 60.1 | 40.9 | **101.7** | 55.6 | 42.9 | 30.5 | 61.1 | 93.2 | 89.3 |
| | 2, 6, 10, 14 | 128 | 37.5 | 69.1 | 74.0 | 61.6 | 59.6 | 38.6 | **101.3** | 57.2 | 43.7 | 36.2 | 75.6 | 99.1 | **98.3** |
| | 3, 7, 11, 15 | 128 | 38.1 | 68.4 | 74.1 | 59.8 | 59.2 | 38.4 | **100.6** | 55.7 | 41.8 | 33.0 | 69.0 | 95.0 | 92.9 |
| FIXED 18% | 1, 6, 11 | 96 | 38.0 | 68.6 | 74.4 | 61.5 | 59.2 | 37.2 | **100.8** | 54.0 | 39.2 | 18.5 | 50.9 | 97.6 | 82.1 |
| | 2, 7, 12 | 96 | 38.6 | 69.4 | 74.4 | 61.6 | 59.4 | 37.0 | **101.3** | 54.1 | 41.0 | 31.2 | 62.2 | 98.6 | 90.5 |
| | 3, 8, 13 | 96 | 37.4 | 68.4 | 73.9 | 60.3 | 59.2 | 37.6 | **100.2** | 53.6 | 40.1 | 28.0 | 68.1 | 99.1 | 91.1 |
| | 4, 9, 14 | 96 | 40.4 | 70.6 | 73.9 | 61.2 | 62.0 | 38.2 | **103.0** | 55.0 | 42.2 | 36.0 | 70.7 | 99.0 | **95.5** |
| ANNEALED 50% | 1,3,5,...,15 | 256 | 38.0 | 69.0 | 74.4 | 61.0 | 60.0 | 37.5 | **101.1** | 58.5 | 44.8 | 36.8 | 77.0 | 99.2 | **99.7** |
| ANNEALED 25% | 1, 5, 9, 13 | 128 | 37.6 | 69.1 | 74.0 | 60.8 | 60.2 | 38.8 | **101.3** | 56.5 | 43.5 | 30.0 | 68.0 | 95.5 | 92.6 |
| ANNEALED 12.5% | 1, 13 | 64 | 37.0 | 68.0 | 73.8 | 59.5 | 58.8 | 35.5 | **99.0** | 52.0 | 38.0 | 20.0 | 55.0 | 92.0 | 81.0 |
| RET. AWARE | - | **10** | 37.6 | 69.0 | 74.6 | 60.5 | 62.0 | 36.8 | **101.3** | 54.2 | 42.1 | 34.4 | 71.1 | 99.0 | **95.0** |

placed.

## 5.1. Distillation Settings

We validate our method by distilling **Llama-3.2-1B-Instruct** and **Qwen-2.5-1.5B-Instruct** into their respective hybrids, HYBRID-LLAMA-1B (following Bick et al. (2025a)) and HYBRID-QWEN-2.5-1.5B.

We adapt the MOHAWK framework (Bick et al., 2024) to train these models. Standard MOHAWK requires a *Matrix Orientation* phase to align the student's randomly initialized matrices with the teacher's feature space. In our case, this step is unnecessary because we initialize the student by directly copying the teacher's most critical attention heads. Since these heads preserve the teacher's internal feature alignment, we can bypass orientation and proceed directly to training:

1. **Hidden-State Alignment:** We train the student's new SSM components to approximate the teacher's block outputs. The retained attention heads act as stable "anchors," ensuring the SSMs learn features compatible with the existing residual stream.

2. **Knowledge Distillation:** We fine-tune the full model end-to-end on a mixture of datasets, minimizing the Cross-Entropy loss between student and teacher logits until convergence.

The complete hyperparameter and implementation details are provided in Section A.1.

## 5.2. Downstream Performance

We begin by evaluating a series of distilled hybrids with varying numbers of retained attention heads. Heads are added in order of importance, ranked by their score on the synthetic key-value retrieval task (Table 5), where lower scores indicate greater contribution to retrieval. As shown in Table 1, even a small number of carefully selected G&A heads enables strong performance across benchmarks—particularly on retrieval-heavy tasks.

Performance improves sharply as the top 10–20 G&A heads are added, after which gains begin to plateau. This confirms that in-context retrieval is bottlenecked by a small subset of attention heads, and that the retrieval-guided distillation captures the necessary functionality with minimal attention overhead.

## 5.3. Comparison with Heuristic Placement Strategies

To contextualize the effectiveness of our retrieval-aware placement, we compare against two recent hybrid distillation frameworks. Unlike our targeted approach, these baselines rely on rigid, periodic heuristics:

1. **Fixed interleaving**. This strategy selects a *fixed* uniform interleaving pattern from the start, retaining Attention only at predetermined layer indices (a global stride with a chosen offset). This is the placement heuristic used in **MOHAWK** (Bick et al., 2024) to place Attention layers.

2. **Annealed interleaving**. This strategy applies an *annealed* replacement schedule under a global stride pat-

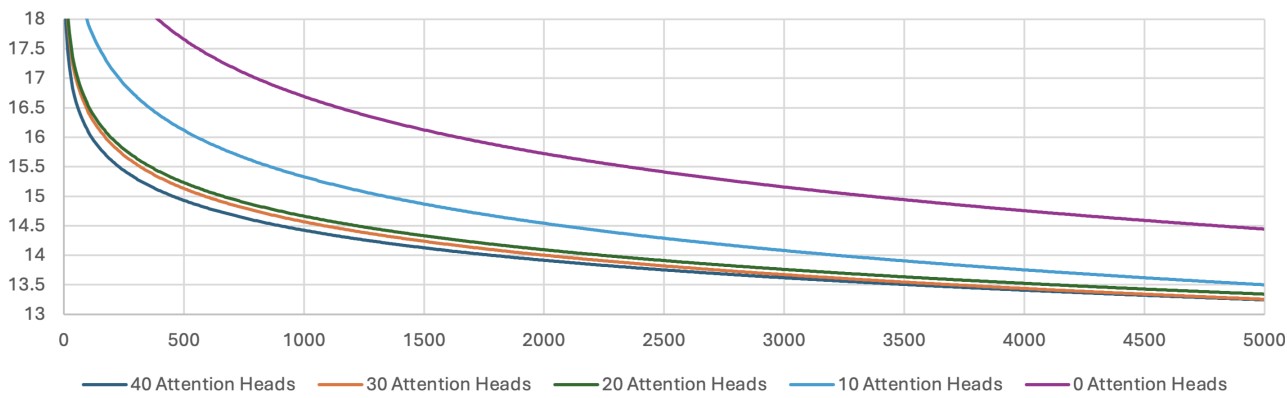

*Figure 3.* **Perplexity during distillation reveals retrieval as the primary bottleneck.** Perplexity is tracked over the first 5,000 training steps for hybrid models with varying numbers of retained G&A heads. Most of the perplexity reduction occurs with the first 10-20 heads—those ranked highest by retrieval importance—with diminishing returns beyond that. This confirms that retrieval is the dominant factor in early language modeling improvements and supports our core claim: targeted retention of retrieval heads is both necessary and sufficient for closing the performance gap.

tern, progressively reducing the fraction of Attention layers across stages (e.g., 50% → 25% → 12.5%). This is the placement heuristic used in **Mamba-In-The-Llama** (Wang et al., 2025b) to place Attention layers. It leverages State-Space-Duality (SSD) (Dao & Gu, 2024) to initialize new SSM parameters $(C, B, X)$ from the teacher Attention weights $(Q, K, V)$; we omit their alignment stages (SFT, DPO) to isolate the effect of attention placement.

Table 3 demonstrates that while all methods achieve teacher-level parity, our approach is significantly more efficient. Where heuristic baselines must retain 25–50% of attention heads to maintain performance, our retrieval-aware method matches this accuracy with only 2% of heads.

This reduction yields substantial hardware benefits. Retrieval-aware distillation reduces memory usage by 3–6× compared to baselines on long sequences (see Section A.4 for detailed calculation). This advantage stems from the fundamental difference in complexity: SSM memory is determined by the state dimension ($O(1)$ w.r.t sequence length), whereas Attention incurs linearly growing KV-cache costs ($O(L)$). By stripping away all attention heads except those strictly necessary for retrieval, we minimize the impact of the memory bandwidth bottleneck (Ivanov et al., 2021; Dao et al., 2022).

### 5.4. Perplexity Reflects Retrieval Capacity

To evaluate whether the retrieval probe selects heads that benefit distillation, we incrementally add attention heads in order of KV-retrieval importance and track validation perplexity during training.

Figure 3 shows a clear pattern: adding the first 10 retrieval-critical heads yields the largest perplexity reduction, and

most gains are reached by 10–20 heads, with diminishing returns thereafter. This supports our ranking strategy, since heads that score highly on synthetic KV-retrieval also drive the largest improvements during distillation, while lower-ranked heads contribute little.

Interpreting absolute perplexity, however, requires care. Some linear baselines (Dao & Gu, 2024; Yang et al., 2024) can achieve lower *overall* perplexity than Transformers, yet Arora et al. (2023) showed that Transformers retain an advantage on tokens that require substantial associative recall. We therefore interpret perplexity changes alongside Table 1: as heads are added, retrieval-heavy benchmarks improve substantially while knowledge-focused tasks change little, suggesting that the perplexity gains primarily reflect better prediction on retrieval-demanding tokens rather than broad improvements in general language modeling.

### 5.5. G&A Functionality Remains Localized in Attention

Bick et al. (2025b) show that in pretrained hybrid models, Gather-and-Aggregate (G&A) roles are naturally delegated to attention heads, due to their ability to attend over the entire sequence. Since our method explicitly preserves the teacher's most retrieval-critical attention heads, we expect this delegation to persist in the distilled hybrid. To verify this, we evaluate the contribution of each head in the hybrid model by ablating them individually and measuring the resulting accuracy drop on the synthetic key-value retrieval task.

Table 6 reports the 20 most impactful heads from both the SSM and attention groups, ranked by retrieval sensitivity. The results reveal a clear divide: **11 out of 20** attention heads fall below the accuracy threshold of 0.7 and are thus considered crucial for retrieval, while only **5 out of 492** SSM heads meet this criterion. This validates our core

*Table 4.* **Reducing state size has minimal impact when retrieval is handled by attention.** Performance of LLAMA-1B with varying SSM state dimensions (`d_state`), keeping the top 10 and 20 retrieval-critical attention heads fixed. Benchmarks follow the setup in Table 1. Reducing `d_state` from 64 to 8 yields minimal performance loss across both knowledge-focused and retrieval-heavy tasks, showing that most retrieval load is offloaded to attention. Further reduction to 4 leads to degradation on retrieval-heavy tasks, suggesting that while attention handles the bulk of retrieval, the SSM still provides auxiliary support.

| # HEADS | STATE SIZE | KNOWLEDGE-FOCUSED | | | | | | | RETRIEVAL-HEAVY | | | | | |
|---|---|---|---|---|---|---|---|---|---|---|---|---|---|---|
| | | ARC-C | ARC-E | PIQA | WG | HS | OBQA | COV | LMB | MMLU | GSM8K | SWDE | KV-Ret | COV |
| 10 | 4 | 36.8 | 68.0 | 74.9 | 60.0 | 60.0 | 36.4 | **99.6** | 50.4 | 36.8 | 26.7 | 69.1 | 70.0 | **79.9** |
| | 8 | 37.3 | 68.5 | 74.5 | 60.3 | 61.0 | 36.4 | **100.8** | 50.4 | 40.0 | 29.5 | 69.5 | 88.0 | **88.1** |
| | 16 | 38.0 | 68.4 | 72.9 | 60.2 | 58.5 | 36.0 | **99.3** | 52.4 | 39.5 | 31.0 | 70.0 | 93.5 | **91.0** |
| | 32 | 37.2 | 68.6 | 74.2 | 60.4 | 59.0 | 36.0 | **100.0** | 53.0 | 41.0 | 32.0 | 70.9 | 97.1 | **93.4** |
| | 64 | 37.6 | 69.0 | 74.6 | 60.5 | 62.0 | 36.8 | **101.3** | 54.2 | 42.1 | 34.4 | 71.1 | 99.0 | **95.0** |
| 20 | 4 | 37.4 | 68.2 | 74.6 | 61.6 | 60.2 | 37.6 | **101.0** | 50.6 | 37.0 | 27.8 | 69.0 | 72.6 | **81.0** |
| | 8 | 38.1 | 69.6 | 74.0 | 61.9 | 61.3 | 38.2 | **102.1** | 51.1 | 41.0 | 30.1 | 71.0 | 90.0 | **90.0** |
| | 16 | 36.8 | 68.2 | 73.6 | 61.4 | 58.2 | 35.4 | **99.3** | 53.2 | 40.3 | 32.1 | 71.0 | 95.4 | **92.1** |
| | 32 | 36.9 | 68.0 | 73.6 | 61.1 | 58.6 | 36.2 | **99.5** | 53.1 | 41.6 | 32.3 | 71.8 | 97.0 | **93.3** |
| | 64 | 38.2 | 69.3 | 74.5 | 62.9 | 61.1 | 36.5 | **101.9** | 55.0 | 43.0 | 34.0 | 72.5 | 99.3 | **95.8** |

assumption: by selectively retaining the teacher's retrieval-critical heads, we ensure that retrieval is handled exactly where it should be—enabling leaner hybrid architectures with interpretable, task-aligned specialization.

### 5.6. Offloading Retrieval Enables Leaner States

State-space models maintain a recurrent state that evolves over time to model long sequences. The dimensionality of this state, `d_state`, controls the model's ability to store and manipulate information across time. We hypothesize that large states are primarily needed for memorization and retrieval. Since SSMs lack access to the full sequence history, they rely on this recurrent memory to carry forward relevant information. However, once retrieval is handled by a small set of explicit G&A attention heads, the recurrent state no longer needs to serve as long-term memory. Instead, it can focus on local language modeling, which requires significantly less capacity.

To test this, we fix the top retrieval-critical attention heads and vary `d_state` across the SSM layers. As shown in Table 4, reducing `d_state` from 64 to 8 preserves strong performance across both knowledge-focused and retrieval-heavy benchmarks. This confirms that once retrieval is offloaded to attention, the SSM backbone can be substantially simplified without degrading effectiveness.

These architectural savings translate directly to inference memory: Table 2 reports per-sequence memory across context lengths for teacher-level hybrids, and shows that pairing a smaller `d_state` with a small set of retained attention heads substantially reduces the footprint across both short and long sequences relative to heuristic baselines. Even with more aggressive compression, coverage remains high: setting `d_state`=4 drops coverage to below 90%, but still substantially exceeds an SSM-only model with `d_state`=64, which reaches only ∼50% coverage

(see Table 1). These results are consistent with Section 5.5, which shows that even a small fraction of SSM heads still contributes to in-context retrieval. This suggests that while attention carries most of the retrieval workload, SSM components still provide auxiliary support for specific retrieval patterns. Fully isolating these roles requires further study.

## 6. Conclusion & Future Work

We introduced *retrieval-aware distillation*, a framework for constructing Transformer–SSM hybrids that preserve performance while reducing reliance on attention. We identify the teacher attention heads responsible for in-context retrieval—those implementing the Gather-and-Aggregate (G&A) mechanism—and retain only these heads in the student. In contrast to prior hybrid distillation approaches that heuristically place attention heads, retrieval-aware distillation directly targets the retrieval mechanism, achieving comparable accuracy with far fewer attention heads and lower KV-cache overhead. We verify the preserved heads maintain their retrieval-related roles and show that, once attention handles retrieval, the SSM backbone can be much leaner, reducing memory with limited performance loss.

Our study leaves several open questions about when retrieval-aware distillation transfers cleanly. First, we only evaluate models below 3B parameters, so it is unclear whether the same small set of heads remains sufficient at larger scale, where retrieval behavior and head redundancy may change. Second, head selection relies on the synthetic KV-retrieval probe of Bick et al. (2025b). While it captures many benchmarks, other retrieval settings (e.g., multi-hop retrieval) may rely on components not identified by this probe. Third, although our hybrid stores fewer total KV pairs than prior hybrids, our distillation does not enforce KV sharing across layers (or within layers via GQA-style tying). Enabling KV sharing during distillation could fur-

ther reduce the KV cache, but we leave this to future work. Finally, retrieval is not fully separated from the recurrent backbone: our ablations show that a small number of SSM heads still affect retrieval tasks. This coupling limits how aggressively we can shrink d_state without degrading retrieval-intensive performance, and achieving a cleaner separation remains an open direction.

Overall, these findings suggest that hybrid designs do not need to retain attention uniformly across layers. Instead, retrieval appears to be driven by a small set of heads, and preserving these heads is sufficient to maintain retrieval behavior while enabling more efficient and compact models.

## Impact Statement

This paper presents work whose goal is to advance the field of Machine Learning. It specifically focuses on linear models and state space models and improve their recall abilities for long sequences. There are many potential societal consequences of our work, none which we feel must be specifically highlighted here.

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

# A. Appendix

## A.1. Experimental Setup

All experiments detailed in Section 5 were conducted using hybrid models trained in mixed precision with Fully Sharded Data Parallel (FSDP) on a single node equipped with 8 NVIDIA H100 GPUs. We utilized activation checkpointing throughout to maximize memory efficiency. Optimization was performed using AdamW with $\beta_1 = 0.9$, $\beta_2 = 0.95$, and a weight decay of 0.1, alongside a global batch size of 128.

For the learning rate schedule, we adopted the Warm-Stable-Decay (WSD) strategy (Hu et al., 2024) with a minimum learning rate of $1 \times 10^{-8}$. The warm-up and decay phases each comprised 10% of the total training steps. Peak learning rates were set to $1 \times 10^{-4}$ for the Hidden-State Alignment stage and $1 \times 10^{-5}$ for Knowledge Distillation.

Throughout all distillation stages, we utilized a balanced 50-50 mixture of Fineweb-edu (Lozhkov et al., 2024) and Finemath-4Plus (Allal et al., 2025), with input sequences packed to a length of 2048. In total, the distillation process consumed 12 billion tokens, demonstrating significant data efficiency compared to full pretraining.

We also found that the effect of distilling from a larger teacher depends on how the student was produced. In particular, Llama-3.2-1B (Touvron et al., 2023) is itself distilled from a larger model, whereas Qwen-2.5-1B was pretrained from scratch. This distinction provides useful guidance and may help explain why teacher size has little effect for Qwen-2.5-1B, but matters for Llama-3.2-1B. Accordingly, for Llama-3.2-1B we repeated the final distillation stage with two teachers—Llama-3.2-1B-Instruct and Llama-3.1-8B-Instruct—as in Bick et al. (2025a).

For the same reason, we also distilled a full Transformer student from a full Transformer teacher, to verify that using the larger teacher does not introduce capabilities beyond those already present in the student's original distillation pipeline.

## A.2. Adapter Implementation

The Adapter is a zero-parameter-overhead mechanism for integrating retained attention heads. It intervenes in the standard attention computation flow **between** the head concatenation and the final output projection. In a standard Transformer, attention heads are concatenated and immediately projected:

$$\text{Output} = \text{Linear}(\text{Concat}(\text{Heads}))$$

In our hybrid architecture, we inject a parameter-free normalization step before this projection to align the attention distribution with the parallel SSM branch:

$$\text{Output} = \text{Linear}(\text{Normalize}(\text{Concat}(\text{Heads}) \mid \text{SSM}_{\text{stats}}))$$

The linear layer (`out_proj`) corresponds to the original attention output projection (simply moved after normalization), thus the Adapter introduces **no additional learnable parameters** compared to the original teacher architecture.

```python
class Adapter(nn.Module):
    def __init__(self, d_model):
        super().__init__()
        # Corresponds to standard Attention o_proj (reused here)
        self.out_proj = nn.Linear(d_model, d_model, bias=True)

    def forward(self, att_hidden, ssm_hidden, hidden_states):
        """att_hidden: [B, L, D] concatenated heads before projection"""
        return self.out_proj(self.normalize(att_hidden, ssm_hidden))

    def normalize(self, x, y, eps=1e-5):
        # 1. Compute stats (x=Attention, y=SSM)
        mu_x, std_x = x.mean(dim=-1, keepdim=True), x.std(dim=-1, keepdim=True)
        mu_y, std_y = y.mean(dim=-1, keepdim=True), y.std(dim=-1, keepdim=True)

        # 2. Whiten Attention, then re-color with SSM stats
        return ((x - mu_x) / (std_x + eps)) * (std_y + eps) + mu_y
```

*Listing 1.* **PyTorch implementation of the Adapter.** The module intervenes between the attention head concatenation and the output projection. It normalizes the attention states using SSM statistics, then applies the standard output projection. Since `out_proj` acts as the block's standard output projection, the net parameter count is unchanged.

## A.3. Retrieval Head Analysis

We conduct a two-stage ablation study to understand where retrieval capabilities reside in both the teacher (Transformer) and the student (Hybrid).

### A.3.1. IDENTIFYING SPECIALISTS IN THE TEACHER

First, we isolate the specific attention heads responsible for retrieval in the teacher model (LLAMA-3.2-1B). By removing heads individually and measuring the drop in KV-Retrieval accuracy, we rank them by importance. As shown in Table 5, the top critical heads are not distributed uniformly; they cluster significantly in the upper layers (e.g., Layers 8–10). This distribution often overlaps with heads identified in prior work as critical for knowledge-heavy tasks like MMLU, suggesting a link between retrieval mechanisms and factual recall.

### A.3.2. VERIFYING DELEGATION IN THE STUDENT

Next, we verify that this functionality persists after distillation into the HYBRID-LLAMA student. In this architecture, we retained the top-20 attention heads found above and converted the remaining 492 into SSM heads. Table 6 validates that the functional role is preserved: when we ablate the student's heads, **11 of the 20** retained attention heads prove critical for retrieval (causing accuracy to drop below 0.7). In contrast, only **5 of the 492** SSM heads show similar sensitivity. This confirms that the G&A (Gather-and-Aggregate) functionality remains successfully localized within the small subset of attention heads, while the SSM components handle the bulk of general sequence modeling.

*Table 5.* **Teacher Ablation (Llama-3.2-1B).** To guide our retention strategy, we rank the teacher's attention heads by their contribution to KV-Retrieval. We list the **top 40 heads** where removal causes the sharpest accuracy drops (lower Acc = higher importance). Note the clustering in upper layers (8, 9, 10), which informs the placement of attention anchors in the hybrid student.

| # | Column A | | | # | Column B | | |
|---|---|---|---|---|---|---|---|
| | **Layer** | **Head** | **Acc** | | **Layer** | **Head** | **Acc** |
| 1 | 9 | 25 | 0.061 | 21 | 8 | 21 | 0.168 |
| 2 | 4 | 4 | 0.092 | 22 | 5 | 27 | 0.169 |
| 3 | 10 | 26 | 0.123 | 23 | 6 | 15 | 0.170 |
| 4 | 0 | 20 | 0.132 | 24 | 15 | 14 | 0.171 |
| 5 | 8 | 20 | 0.132 | 25 | 8 | 30 | 0.173 |
| 6 | 8 | 19 | 0.144 | 26 | 9 | 12 | 0.173 |
| 7 | 8 | 16 | 0.146 | 27 | 3 | 8 | 0.175 |
| 8 | 7 | 22 | 0.148 | 28 | 4 | 6 | 0.175 |
| 9 | 10 | 20 | 0.148 | 29 | 5 | 4 | 0.175 |
| 10 | 5 | 5 | 0.150 | 30 | 0 | 11 | 0.177 |
| 11 | 7 | 13 | 0.153 | 31 | 3 | 15 | 0.178 |
| 12 | 11 | 8 | 0.153 | 32 | 5 | 1 | 0.178 |
| 13 | 1 | 28 | 0.154 | 33 | 5 | 13 | 0.179 |
| 14 | 4 | 26 | 0.157 | 34 | 4 | 11 | 0.180 |
| 15 | 5 | 18 | 0.159 | 35 | 12 | 3 | 0.180 |
| 16 | 5 | 31 | 0.159 | 36 | 0 | 3 | 0.181 |
| 17 | 0 | 7 | 0.162 | 37 | 3 | 16 | 0.181 |
| 18 | 3 | 29 | 0.165 | 38 | 5 | 17 | 0.181 |
| 19 | 10 | 9 | 0.166 | 39 | 7 | 24 | 0.181 |
| 20 | 6 | 31 | 0.168 | 40 | 8 | 5 | 0.181 |

*Table 6.* **Student Ablation (Hybrid-Llama).** We validate the student model (20 Attn, 492 SSM heads) by ablating components individually. We report the most sensitive heads from each group. **Bold** values indicate a drop below 0.7 accuracy. Results show 11/20 attention heads are critical vs. only 5/492 SSM heads, confirming that retrieval is effectively offloaded to attention.

| # | SSM Heads | | | Attn Heads | | |
|---|---|---|---|---|---|---|
| | **Layer** | **Head** | **Acc** | **Layer** | **Head** | **Acc** |
| 1 | 9 | 22 | **0.016** | 8 | 0 | **0.001** |
| 2 | 1 | 1 | **0.504** | 8 | 1 | **0.005** |
| 3 | 4 | 15 | **0.561** | 0 | 0 | **0.007** |
| 4 | 9 | 9 | **0.632** | 10 | 1 | **0.028** |
| 5 | 7 | 13 | **0.673** | 7 | 1 | **0.093** |
| 6 | 8 | 1 | 0.723 | 9 | 0 | **0.333** |
| 7 | 2 | 17 | 0.734 | 8 | 2 | **0.360** |
| 8 | 8 | 18 | 0.749 | 10 | 0 | **0.589** |
| 9 | 0 | 21 | 0.750 | 5 | 0 | **0.642** |
| 10 | 10 | 7 | 0.758 | 5 | 2 | **0.663** |
| 11 | 6 | 18 | 0.761 | 1 | 0 | **0.689** |
| 12 | 4 | 18 | 0.762 | 7 | 0 | 0.779 |
| 13 | 4 | 0 | 0.763 | 3 | 0 | 0.791 |
| 14 | 8 | 31 | 0.765 | 4 | 0 | 0.798 |
| 15 | 1 | 3 | 0.766 | 4 | 1 | 0.806 |
| 16 | 0 | 2 | 0.767 | 5 | 1 | 0.814 |
| 17 | 4 | 20 | 0.767 | 10 | 2 | 0.818 |
| 18 | 6 | 9 | 0.768 | 11 | 0 | 0.820 |
| 19 | 1 | 8 | 0.771 | 6 | 0 | 0.890 |
| 20 | 5 | 9 | 0.774 | 0 | 1 | 0.891 |

## A.4. Memory Limits

Modern inference workloads are increasingly bottlenecked by memory traffic rather than compute. To analyze this, we break down memory usage into two distinct categories:

- **State Memory (Constant):** Fixed-size hidden states maintained by SSM heads. This cost is paid upfront regardless of sequence length ($L$).

- **KV-Cache Memory (Linear):** Key-Value pairs stored by Attention heads for every token. This cost grows linearly with $L$.

We define the total memory footprint $M(L)$ for a sequence of length $L$ as:

$$M(L) = \underbrace{(N_{\text{SSM}} \times d_{\text{head}} \times d_{\text{state}} \times 2)}_{\text{State Memory (Bytes)}} + \underbrace{(N_{\text{KV}} \times d_{\text{head}} \times 2 \times 2 \times L)}_{\text{KV Memory (Bytes)}}$$

where we assume `bfloat16` precision (2 bytes/element) and head dimension $d_{\text{head}} = 64$. Here $N_{\text{SSM}}$ is the total number of SSM heads across layers. For attention, $N_{\text{KV}}$ is the number of *KV heads per layer* (it is *not* summed over layers): under GQA, $N_{\text{KV}} = N_{\text{attn}}/g$, where $N_{\text{attn}}$ is the number of query heads and $g$ is the group size. For example, $N_{\text{attn}} = 128$ with $g = 4$ gives $N_{\text{KV}} = 128/4 = 32$.

**Example Calculation: Retrieval-Aware model.** Our hybrid model replaces most attention heads with Mamba heads to reduce the linear growth factor. Summing across all layers, we utilize 492 SSM heads (state dim 8) and 10 Attention heads:

$$\text{State} = 492 \times 64 \times 8 \times 2 \text{ bytes} \approx \textbf{492} \text{ KB}$$
$$\text{KV} = 10 \times 64 \times 2 \times 2 \times L \text{ bytes} \approx \textbf{2.5L} \text{ KB}$$

Thus, $M_{\text{HL}}(L) \approx 492 + 2.5L$ (KB).

Table 7 extends this breakdown to all baselines.

*Table 7.* **Memory Bottleneck Analysis.** Detailed breakdown of the fixed State Memory versus the linear KV-Cache component. Counts ($N$) represent totals across all layers. Values are converted to **KB** (1024 bytes) for readability. **Total Footprint** sums these to define the memory usage function $M(L)$.

| Model | State Memory (Fixed) | | KV-Cache (Linear) | | Total Footprint $M(L)$ |
|---|---|---|---|---|---|
| | *Calc. (Elements)* $(N_{\text{SSM}} \times d_H \times d_S)$ | *Size (KB)* | *Calc. (Elements)* $(N_{\text{KV}} \times d_H \times 2 \times L)$ | *Size (KB)* | *Formula* $(A + B \cdot L)$ (KB) |
| RETRIEVAL-AWARE | $492 \times 64 \times 8$ | **492** | $10 \times 64 \times 2 \times L$ | **2.5 L** | $492 + 2.5L$ |
| LAYER-WISE (25%) | $384 \times 64 \times 64$ | 3,072 | $32 \times 64 \times 2 \times L$ | 8 L | $3{,}072 + 8L$ |
| LAYER-WISE (50%) | $256 \times 64 \times 64$ | 2,048 | $64 \times 64 \times 2 \times L$ | 16 L | $2{,}048 + 16L$ |
| LLAMA-3.2-1B | — | 0 | $128 \times 64 \times 2 \times L$ | 32 L | $32L$ |

