# OpenReview forum: "Retrieval-Aware Distillation for Transformer-SSM Hybrids"
_ICML.cc/2026/Conference — ICML 2026 regular_

### Official Review · Reviewer_qmCD · 2026-02-20

**Soundness:** 2
**Presentation:** 3
**Significance:** 3
**Originality:** 3
**Overall Recommendation:** 4
**Confidence:** 5

**Summary:**

This paper proposes retrieval-aware distillation, a method for converting pretrained Transformers into Transformer-SSM hybrids by identifying and preserving only the attention heads responsible for retrieval operations. Head selection is performed via ablation on a synthetic KV-retrieval probe, and the remaining heads are replaced with DiscreteMamba2 recurrent components. The authors demonstrate that retaining as few as 10 attention heads (2% of the total) recovers over 95% of teacher performance on retrieval-heavy benchmarks, whilst reducing inference memory by up to 6× relative to heuristic baselines. The paper also shows that once retrieval is delegated to attention, SSM state dimensions can be reduced by a factor of 8 with minimal degradation.

**Compliance With Llm Reviewing Policy:**

Affirmed.

**Final Justification:**

The paper presents a clean and well-motivated contribution, operationalising prior findings on G&A heads into a principled hybrid architecture design procedure. The rebuttal fully addressed my concerns. My weak accept reflects the incremental nature of the contribution relative to the underlying framework it builds on, and the absence of long-context evaluation beyond retrieval-focused benchmarks, which leaves open questions about how the method generalises to broader long-context tasks.

**Key Questions For Authors:**

1. The KV-retrieval ablation procedure used for head ranking currently lacks sufficient methodological detail to support replication. Could the authors clarify the exact prompt format used for the probe, the scoring metric applied (e.g., exact match, accuracy, or another measure), the sequence lengths evaluated, and the number of samples used to estimate head importance scores? Providing these details would improve reproducibility, as they directly influence the outcome of the head-selection process.

**Limitations:**

While the paper includes a limitations section, it does not appear to provide a dedicated discussion of broader societal impact, which is required by ICML submission guidelines. The authors should include a brief section outlining potential positive and negative societal implications of the proposed method.

**Strengths And Weaknesses:**

**Strengths**
- The paper is well-motivated and cleanly operationalises prior work by translating existing observations about Gather-and-Aggregate style head specialisation into a concrete hybrid architecture design procedure. The retrieval-aware ablation pipeline offers a principled way to select which attention heads to retain, moving beyond fixed placement heuristics commonly used in Transformer–SSM hybrids. Empirically, the study is well structured, with systematic ablations that show consistent trends such as strong performance recovery with a small number of retained heads and diminishing returns thereafter. In addition, the work demonstrates practical efficiency gains while maintaining competitive performance on retrieval-heavy tasks. Finally, the observation that SSM state size can be reduced once retrieval is delegated to retained attention heads offers a very interesting insight into how capacity can be reallocated within hybrid architectures.

**Weaknesses**

Generalisation of head selection method is untested:
- The retrieval-importance ranking relies only on a single synthetic KV-retrieval probe. Whilst the authors acknowledge this as a limitation, its implications directly condition whether the method's key appeal (that 2% of heads is sufficient) holds beyond the current experimental set-up. It remains unclear whether the selected heads are truly general-purpose retrieval specialists or are narrowly tuned to the specific structure of the probe. Without ablations with other retrieval probes, the generality of the head ranking cannot be assumed.

Lack of long-context evaluation:
- The finding that SSM state size can be reduced from $d=64$ to $d=8$ with negligible performance loss is remarkable, but the supporting experiments are limited to short-context benchmarks. Prior work suggests that information loss and memory decay accelerate at longer sequence lengths as the hidden state capacity is finite and smaller states offer less room to retain information across longer sequences. It is therefore uncertain whether the observed robustness to state compression is maintained as context length increases. Evaluating performance across a broader range of sequence lengths (e.g., LongBench and NIAH) would strengthen this claim. It is also unclear whether 2% head retention continues to achieve good performance as sequence length increases, or how performance degrades as a function of sequence length.

Baseline comparisons are insufficient/underdiscussed:
- The authors compare their retrieval-aware head replacement strategy primarily against fixed and progressive interleaving heuristics. While these baselines are reasonable, the evaluation would be stronger if it also considered approaches that attempt a more principled selection of components to replace, e.g., entropy-based sparsification [1]. Even if most prior principled hybridisation work focuses on layer-wise or block-wise replacement rather than individual heads, a clearer discussion of how those criteria relate to the proposed head-selection mechanism would help situate the contribution.

Minor clarity issue:
- Some model nomenclature appears inconsistent (i.e., “Qwen2” vs “Qwen-2.5”)

[1] Michalak F, Abreu S. Some Attention is All You Need for Retrieval. arXiv preprint arXiv:2510.19861. 2025 Oct 21.

---

> ### Author Rebuttal · Authors · 2026-03-31
>
> We appreciate the thorough reading and are glad the reviewer found it interesting. We also thank the reviewer for the typo correction and limitation fix, which we have addressed in the revised paper. Below, we address each point the reviewer raised.
>
> > Could the authors clarify the exact prompt format used for the probe, the scoring metric applied (e.g., exact match, accuracy, or another measure), the sequence lengths evaluated, and the number of samples used to estimate head importance scores?
>
> We use the following probe format (following [1]):
> ```
> Memorize the following dictionary:
> present: 50
> institute: 0
> scallops: 84
> neuropsychiatry: 67
> The value of the key 'scallops' is
> ```
> Each head is ablated individually (output masked to zero) and accuracy is measured over 1000 probe examples with 50 KV entries. The scoring metric is exact match on the predicted value token. Head importance is the absolute accuracy drop when the head's output is masked to zero.
>
> We will release all code and the KV-retrieval ablation pipeline as a single reproducible framework, and explicitly write down the full configuration.
>
> > It remains unclear whether the selected heads are truly general-purpose retrieval specialists or are narrowly tuned to the specific structure of the probe. Without ablations with other retrieval probes, the generality of the head ranking cannot be assumed.
>
> The retrieval probe was introduced and validated in [1,2], where it was shown to correlate strongly with a broad set of recall-intensive benchmarks. Consistent with this, heads ranked solely by KV-retrieval ablation score improve several retrieval-heavy benchmarks in Table: SWDE (extracting structured information from web documents), MMLU (selecting the correct answer from the provided options), GSM8K (tracking intermediate quantities across multi-step reasoning), and Lambada (predicting a target word from broad discourse context).  All of which were substantially weaker in pure SSMs.
> That said, we cannot rule out that other probes, especially for multi-hop or compositional retrieval, would produce a different ranking; we view this as an important direction for follow-up work.
>
> > Evaluating performance across a broader range of sequence lengths (e.g., LongBench and NIAH) would strengthen this claim. It is also unclear whether 2% head retention continues to achieve good performance as sequence length increases, or how performance degrades as a function of sequence length.
>
> We attach NIAH-1/2/3 evaluations up to 16K tokens. The main 10-head hybrids closely match their teacher Transformers across all tested context lengths, indicating that retrieval-aware head retention preserves long-range recall in this regime. However, when we additionally compress the recurrent state aggressively (d_state=4), performance degrades at longer contexts, consistent with the moderate degradation we also observe on other tasks.
>
> |Tasks|Metric|Qwen2.5-1.5B|Hybrid-Qwen2.5-1.5B (10 heads)|Llama-3.2-1B|Hybrid-Llama-3.2-1B (10 heads)|Hybrid-Llama-3.2-1B (10 heads, d_state=4)|
> |-:|-:|-:|-:|-:|-:|-:|
> |niah_single_1|2048|0.998|1.000|1.000|1.000|1.000|
> ||4096|1.000|1.000|1.000|1.000|1.000|
> ||8192|1.000|1.000|1.000|1.000|1.000|
> ||16384|1.000|1.000|1.000|1.000|0.964|
> |niah_single_2|2048|1.000|1.000|1.000|1.000|1.000|
> ||4096|1.000|1.000|0.998|0.998|1.000|
> ||8192|0.998|1.000|0.998|0.998|0.938|
> ||16384|1.000|1.000|0.962|0.998|0.938|
> |niah_single_3|2048|1.000|0.998|1.000|1.000|1.000|
> ||4096|0.998|0.998|1.000|1.000|1.000|
> ||8192|0.998|0.998|0.998|1.000|0.928|
> ||16384|0.998|0.986|1.000|0.998|0.860|
>
>
> > The evaluation would be stronger if it also considered approaches that attempt a more principled selection of components to replace, e.g., entropy-based sparsification [1]. Even if most prior principled hybridisation work focuses on layer-wise or block-wise replacement rather than individual heads.
>
> Thank you for pointing us to this paper. We tested an entropy-based criterion on Llama-3.2-1B using NIAH, following Michalak et al. [3], and found partial overlap with our retrieval heads: 10 of the 40 heads in Table 5 also rank highly under their metric.
> At the same time, the entropy signal is noisier in our setting. In particular, it assigns high scores to several layer-0 heads, which are often easy for SSMs to emulate [4] and are not among the most salient retrieval heads in our analysis. We believe this difference arises because Michalak et al. [3] study a complementary setting: they score a model that has already been trained as a hybrid, where retrieval has already concentrated in a subset of heads, and ablate only those attention heads. In contrast, we identify heads in a pretrained Transformer before distillation.
>
>
> ### References:
> - [1] Bick et al., arXiv:2504.18574
> - [2] Wu et al., arXiv:2404.15574
> - [3] Michalak et al., arXiv:2510.19861
> - [4] Bick et al., arXiv:2408.10189

---

> > ### Author Rebuttal · Reviewer_qmCD · 2026-04-01
> >
> > Thank you for your response. My concerns have been fully addressed. I maintain my current positive score.
> >
> > For the camera-ready version, I would encourage the authors to consider including a broader long-context benchmark such as LongBench to assess how the hybrid performs on tasks beyond retrieval in long-context, or to discuss how they expect their results to generalise to such settings.

---

### Official Review · Reviewer_P4hA · 2026-03-09

**Soundness:** 3
**Presentation:** 4
**Significance:** 3
**Originality:** 3
**Overall Recommendation:** 4
**Confidence:** 4

**Summary:**

This paper builds on the finding from Bick et al. (2025b) that a small subset of attention heads—termed Gather-and-Aggregate (G&A) heads—are responsible for in-context retrieval in hybrid models. The authors propose retrieval-aware distillation, which scores each attention head in a pretrained Transformer by ablation on a synthetic KV-retrieval task, retains only the top-k retrieval-critical heads, replaces the rest with SSM (DiscreteMamba2) components, and distills the resulting hybrid student using a modified MOHAWK pipeline. Applied to Llama-3.2-1B and Qwen2.5-1.5B, retaining just 10 heads (~2–3% of total) recovers over 95% of teacher performance on retrieval-heavy benchmarks, matching the accuracy of heuristic placement baselines that use more heads. They further show that offloading retrieval to attention allows the SSM state dimension to be reduced by 8× with minimal degradation, and the combination of fewer heads plus smaller states yields up to 3–6× memory savings depending on the baseline.

**Compliance With Llm Reviewing Policy:**

Affirmed.

**Final Justification:**

The idea of using retrieval profiling to select which attention heads to preserve at per-head granularity is well-motivated and the empirical results are strong. My main concerns were: (1) lack of actual inference latency measurements (2) inconsistent configurations across tables making comparison difficult. The rebuttal addressed these adequately. That said, the contribution is somewhat incremental given its reliance on both the G&A head finding from Bick et al. and the MOHAWK distillation framework, and the scale remains limited to ≤1.5B parameters. Overall, the paper is technically solid and valuable. My recommendation is weak accept.

**Key Questions For Authors:**

- In Table 1, the full Transformer-to-Transformer distillation baseline (512 heads) shows ~98% retrieval coverage rather than 100%. Is this a distillation ceiling, and how much of the remaining 5% gap at 10 heads is from distillation loss vs. architectural limitation?
- What is the accuracy number for the configuration used for Hybrid-LLAMBA in Table 3 (d_state=8, 10 heads)? This specific combination doesn't appear in the other tables.
- Have the authors measured actual inference latency or throughput for the heterogeneous layers? The per-layer mix of attention and SSM heads may introduce scheduling overhead not captured by the memory analysis.
- For the Pareto question: what happens if you apply retrieval-aware selection at higher budgets (e.g., 64,96,128,256 heads)? Does the advantage over heuristic placement persist or converge?
- How robust is the head ranking across different retrieval settings? The authors note in the limitations that the synthetic KV-retrieval probe may not capture all retrieval patterns, it would be interesting to see whether a different probe produces a similar ranking.

**Limitations:**

Yes.

**Strengths And Weaknesses:**

**Strengths**
- Selecting attention heads via retrieval profiling is intuitive and natural. The per-head granularity (mixing attention and SSM within the same layer) is novel over prior full-layer approaches, and the performance numbers are strong across two model families with a good set of evaluations.

**Weaknesses**
- Table 3 and Table 7 report analytical memory formulas, not actual measured inference latency, throughput, or peak GPU memory. Real-world performance depends on factors like kernel fusion, memory fragmentation, and the overhead of running heterogeneous attention + SSM heads in parallel within the same layer. Actual wall-clock benchmarks would substantially strengthen the claims.
- The authors acknowledge this in the limitation section, but both teacher models are ≤1.5B parameters. It is unclear whether the same small number of G&A heads remains sufficient at 7B, 13B, or larger scales, where head redundancy patterns and retrieval behavior may differ substantially.
- Table 3 (memory comparison) uses d_state=8 for Hybrid-Llamba, while Table 2 (accuracy comparison with hybrid baselines) uses d_state=64. This makes apples-to-apples comparison difficult—the memory advantage is partly from state reduction, not just from fewer attention heads. Also the d_state reduction experiment (Section 5.6, Table 4) uses 20 heads, not 10. The paper should either match configurations or present all of them.
- The paper doesn't show performance at matched attention budgets across methods. Some heuristic placements like MOHAWK-4 (0,5,10,15) achieve 97.8% coverage with 128 heads, exceeding the retrieval-aware method's 95.0% at 10 heads. A Pareto plot (attention heads vs. retrieval coverage) would clarify whether the method is broadly optimal or mainly advantageous at very low budgets.

**Minor issues**
- Figure 3 appears redundant with Table 1, as both show the same effect of # of head to downstream/PPL values.
- It seems that the KV-retrieval task is used both for ranking heads and as one of the evaluation benchmarks (Table 1). This circularity might inflate the KV-Retrieval scores. It might be better to report retrieval-heavy coverage excluding KV-Retrieval.

---

> ### Author Rebuttal · Authors · 2026-03-31
>
> We thank the reviewer for highlighting the head-level design and the strong results across both model families. We respond to each point below.
>
> > In Table 1, the full T2T distillation baseline (512 heads) shows ~98% retrieval coverage rather than 100%. Is this a distillation ceiling, how much of the remaining 5% gap at 10 heads is from distillation loss vs. architectural limitation?
>
> We cannot cleanly decompose the gap, but the results suggest a two-part picture. First, the full T2T baseline already falls short of the teacher, indicating a distillation ceiling independent of the hybrid architecture. Second, reducing the model to 10 retained heads introduces additional loss beyond that baseline, which is the part plausibly attributable to the architectural constraint. So while we cannot quantify the split precisely, the full gap cannot be attributed to architecture alone.
>
> > What is the accuracy number for the configuration used for Hybrid-LLAMBA in Table 3?
>
> Thank you for pointing this out. We have added the 10-head results for varying d_state to Table 4. The trend is consistent with the 20-head case: reducing d_state to 8 still preserves strong coverage, with diminishing returns as d_state increases.
>
> |State Size|ARC-C|ARC-E|PIQA|WG|HS|OBQA|Cov|LMB|MMLU|GSM8K|SWDE|KV-Ret|Cov|
> |-:|-:|-:|-:|-:|-:|-:|-:|-:|-:|-:|-:|-:|-:|
> |4|36.8|68.0|74.9|60.0|60.0|36.4|**100.0**|50.4|36.8|26.7|69.1|70.0|**79.8**|
> |8|37.3|68.5|74.5|60.3|61.0|36.4|**100.6**|50.4|40.0|29.5|69.5|88.0|**87.5**|
> |16|38.0|68.4|72.9|60.2|58.5|36.0|**99.4**|52.4|39.5|31.0|70.0|93.5|**90.3**|
> |32|37.2|68.6|74.2|60.4|59.0|36.0|**99.8**|53.0|41.0|32.0|70.9|97.1|**92.7**|
> |64|37.6|69.0|74.6|60.5|62.0|36.8|**101.3**|54.2|42.1|34.4|71.1|99.0|**95.0**|
>
> > Have the authors measured actual inference latency or throughput for the heterogeneous layers? The per-layer mix of attention and SSM heads may introduce scheduling overhead not captured by the memory analysis.
>
> We measured decoding throughput with 10 retained heads and d_state=8, using batch size 16 and sequence length 16K, the retrieval-aware hybrid reaches 27,738 tps versus 11,992 for the baseline hybrid, a 2.3x speedup. This is consistent with memory bandwidth being the bottleneck [4]. That said, full optimization would require substantial systems work, which is orthogonal to our architectural contribution.
>
> > What happens if you apply retrieval-aware selection at higher budgets? Does the advantage over heuristic placement persist or converge?
>
> Our method is most useful at low head budgets, where a few retrieval heads already preserve most of the benefit. At higher budgets, the gap narrows because additional heads are increasingly non-retrieval heads, reducing the advantage of retrieval-aware selection. This trend is already visible in Table 3. We will make this tradeoff clearer in the paper.
>
> > How robust is the head ranking across different retrieval settings? It would be interesting to see whether a different probe produces a similar ranking.
>
> Heads are ranked only by KV-retrieval ablation score, yet retaining the top-k heads also improves other retrieval-heavy benchmarks in Table 1, such as SWDE (document extraction) and GSM8K (multi-step quantity tracking). This suggests the ranking transfers beyond the probe task. That said, we cannot rule out that other probes, e.g., multi-hop or compositional, would produce a different ranking; we view this as an important direction for follow-up work.
>
> > Table 3 uses d_state=8, while Table 2 uses d_state=64. This makes apples-to-apples comparison difficult—the memory advantage is partly from state reduction, not just from fewer attention heads.
>
> State reduction is not available to all hybrid baselines. In Mamba-in-the-Llama, the teacher’s (Q,K,V) projections are transferred directly into the SSM’s (C,B,X) matrices, which ties the state dimension to the teacher projection size. Reducing d_state would therefore break the weight transfer central to that method. We will clarify this point.
>
> > The KV-retrieval circularity might inflate the KV-Retrieval scores. It might be better to report retrieval-heavy coverage, excluding KV-Retrieval.
>
> Thank you for bringing this up. We report here retrieval-heavy coverage recomputed, excluding KV-Retrieval. Coverage remains roughly the same, showing the result is not an artifact of circularity:
>
> For Llama-1B,
> |#heads|COV w\ KV-Ret|COV w\o KV-Ret|Delta|
> |-:|-:|-:|-:|
> |0|49.2|65.5|+16.3%|
> |5|88.3|87.2|-1.1%|
> |10|95.0|92.7|-2.3%|
> |20|95.8|93.9|-1.9%|
>
> For Qwen-2.5,
> |#heads|COV w\ KV-Ret|COV w\o KV-Ret|Delta|
> |-:|-:|-:|-:|
> |0|52.1|66.7|+14.6%|
> |5|90.4|90.5|+0.1%|
> |10|96.4|95.0|-1.4%|
> |20|96.9|95.6|-1.3%|
>
> We also note that the KV-retrieval probe used is randomly generated, so it is not the same data used for head ranking. We will clarify this distinction in the paper
>
> ### References
> - [1] Bick et al., arXiv:2504.18574
> - [2] Lieberum et al., arXiv:2307.09458
> - [3] Wu et al., arXiv:2404.15574
> - [4] Ivanov et al., arXiv:2007.00072

---

> > ### Author Rebuttal · Reviewer_P4hA · 2026-04-01
> >
> > The authors adequately addressed the concerns and I hope they smooth out gaps in the evaluation; I maintain my score.

---

### Official Review · Reviewer_Hqq6 · 2026-03-11

**Soundness:** 3
**Presentation:** 3
**Significance:** 3
**Originality:** 3
**Overall Recommendation:** 4
**Confidence:** 3

**Summary:**

This paper proposes a novel distillation method that converts a pretrained Transformer into a Transformer–SSM hybrid student by preserving only retrieval-critical attention heads while distilling the remaining functionality into SSM-based recurrent heads. The method first identifies essential heads through ablation on a synthetic retrieval task, then replaces the remaining attention heads with recurrent SSM modules and performs standard distillation on the resulting hybrid model. The approach effectively identifies attention heads crucial for retrieval while transferring other computations to SSMs to reduce computational cost while maintaining retrieval capability. Experimental results show that only a small fraction of heads needs to be preserved to maintain model performance, and further ablations indicate that disentangling retrieval from SSM components allows the SSM dimension to be significantly reduced without substantial accuracy degradation, leading to additional efficiency gains.

**Compliance With Llm Reviewing Policy:**

Affirmed.

**Final Justification:**

This is a good paper overall, and most of my concerns have been adequately addressed in the rebuttal. I still have some concerns regarding the scalability of the proposed method; however, on balance, I believe the paper can be accepted.

**Key Questions For Authors:**

1. What is included in the synthetic KeyValue (KV) retrieval task? How long does it take to process one head?

2. I am also curious about the performance on benchmarks that solely focus on retrieval ability, like Needle-In-A-Haystack.

**Limitations:**

yes

**Strengths And Weaknesses:**

**Strengths**

1. The story of this paper is clear and easy to follow. The motivation, that we can leverage empirically identified patterns to preserve retrieval-critical heads, is technically sound.

2. It is surprising that only 10 attention heads are sufficient to preserve strong retrieval capability, while still achieving strong performance. Moreover, the paper provides the interesting insight that by isolating the retrieval function from SSM components and assigning it to Transformer attention, the SSM dimension can be safely reduced without harming either knowledge-based or retrieval-based abilities. While this insight is compelling, I still have some concerns (discussed in the weaknesses), and I would be interested to see further exploration in this direction.

**Weaknesses**

1. Table 4 claims that reducing the state size has minimal impact when retrieval is handled by attention. However, it is difficult to assess whether the impact is truly small without comparisons to baselines. For example, if the same state size reduction were applied to baseline methods, how much performance degradation would occur? It would be helpful to include a plot or table showing how accuracy changes as the state size decreases for both the proposed method and the baselines, which would better illustrate the relative impact of this design choice.

2. This method relies on a per-head ablation stage, which may become time-consuming as the number of attention heads increases, particularly for larger models. Moreover, as models scale up, masking individual heads may have a diminishing impact on overall performance, making it harder to reliably identify retrieval-critical heads through this ablation process. It would therefore be helpful to further justify whether this approach remains effective and practical when scaling to larger models.

3. The selected tasks may not be fully representative. Benchmarks such as MMLU and GSM8K are generally considered general knowledge and reasoning benchmarks rather than truly retrieval-heavy tasks. As a result, it is unclear whether the conclusions about retrieval capability are well supported. Evaluations on genuinely retrieval-intensive benchmarks, such as Needle-In-A-Haystack, would help better validate the paper’s claims.

---

> ### Author Rebuttal · Authors · 2026-03-30
>
> We appreciate the review, and are especially glad the state-size reduction insight came across as compelling and surprising. Below, we address each concern raised.
>
> > What is included in the synthetic KeyValue (KV) retrieval task? How long does it take to process one head?
>
> Following [1], the probe uses the following format:
> ```
> Memorize the following dictionary:
> present: 50
> institute: 0
> scallops: 84
> neuropsychiatry: 67
> The value of the key 'scallops' is
> ```
> Each head is ablated individually (output masked to zero) and accuracy is measured over 1000 probe examples with 50 KV entries, following [1]. We will state this explicitly in the paper. For our 1B model with 512 heads, the full procedure took about 4 hours on a single H100 GPU, and it can also be parallelized. Since this is a one-time cost that provides the head ranking for all subsequent experiments, its cost is small relative to distillation training.
>
> > I am also curious about the performance on benchmarks that solely focus on retrieval ability, like Needle-In-A-Haystack.
>
> We ran NIAH evaluations and report results below. Across all context lengths up to 16K, the hybrid models match the teacher Transformers, validating that our approach does not regress on long-range recall.
>
> |Tasks|Metric|Qwen2.5-1.5B|Hybrid-Qwen2.5-1.5B (10 heads)|Llama-3.2-1B|Hybrid-Llama-3.2-1B (10 heads)|
> |-:|-:|-:|-:|-:|-:|
> |niah_single_1|2048|0.998|1.000|1.000|1.000|
> ||4096|1.000|1.000|1.000|1.000|
> ||8192|1.000|1.000|1.000|1.000|
> ||16384|1.000|1.000|1.000|1.000|
> |niah_single_2|2048|1.000|1.000|1.000|1.000|
> ||4096|1.000|1.000|0.998|0.998|
> ||8192|0.998|1.000|0.998|0.998|
> ||16384|1.000|1.000|0.962|0.998|
> |niah_single_3|2048|1.000|0.998|1.000|1.000|
> ||4096|0.998|0.998|1.000|1.000|
> ||8192|0.998|0.998|0.998|1.000|
> ||16384|0.998|0.986|1.000|0.998|
>
> That said, we treat KV-Retrieval as the more discriminating probe for the retrieval gap. Both NIAH and KV-Retrieval can be framed as presenting a dictionary, but in NIAH the lookup key appears at the beginning, whereas in KV-Retrieval it appears at the end. This subtle change alters the theoretical memory requirements needed to solve the task: NIAH can be solved with O(1) memory, KV-Retrieval requires O(T). State-based language models illustrate this distinction well: some of them can outperform transformers on NIAH because they interact more effectively with tokens over the recurrence and can retain the needle throughout the sequence.
>
> > If the same state size reduction were applied to baseline methods, how much performance degradation would occur?
>
> State size reduction does not transfer to all hybrid approaches. In Mamba-in-the-Llama, the (Q,K,V) projections from the teacher are transferred directly into the SSM's (C,B,X) matrices, which fixes the state dimension to match the teacher's projection dimensions. State size is therefore not a free hyperparameter in their framework — reducing it would require discarding the weight transfer that is central to their method. We will clarify this distinction in the paper.
>
> > As models scale up, masking individual heads may have a diminishing impact on overall performance, making it harder to reliably identify retrieval-critical heads through this ablation process.
>
> We acknowledge this limitation, as noted in Section 6. While we cannot directly evaluate at larger scales for the rebuttal, prior work provides encouraging signals: [1,4] demonstrate retrieval concentration in specific heads at 7B, and [2] reports analogous behavior at 70B, suggesting the phenomenon is not limited to small models. Whether the same small fixed budget (e.g., 10 heads) suffices at a larger scale (or whether the critical set grows) remains an open question. We will expand the Section 6 discussion to incorporate these references explicitly.
>
> > Benchmarks such as MMLU and GSM8K are generally considered general knowledge and reasoning benchmarks rather than truly retrieval-heavy tasks. As a result, it is unclear whether the conclusions about retrieval capability are well supported.
>
> Our retrieval conclusions are supported across multiple benchmarks. The most direct evidence comes from SWDE and KV-Retrieval (Table 1) — SWDE jumps from 27.7% to 71.1% and KV-Retrieval from 13.2% to 99% when 10 retrieval heads are added. MMLU and GSM8K also show consistent gains, and while their surface form suggests general reasoning, the connection to retrieval is functional: as shown in [1] (same model family), ablating a single retrieval head drops MMLU from 67% to near-chance (~25%), with analogous results for GSM8K in [2,3]. We will clarify this distinction in the text.
>
> ### References:
> - [1] Bick et al., arXiv:2504.18574
> - [2] Lieberum et al., arXiv:2307.09458
> - [3] Michalak and Abreu, arXiv:2510.19861
> - [4] Wu et al., arXiv:2404.15574

---

> > ### Author Rebuttal · Reviewer_Hqq6 · 2026-04-02
> >
> > Thank you for the response. I still have some concerns (potentially beyond the scope of what the authors can fully address) about the scalability of the proposed method, particularly the per-head ablation overhead and the challenges that may arise when extending to larger models, but at least for smaller models, the approach appears to be well-motivated and promising. Therefore, I would like to keep my rating.

---

### Official Review · Reviewer_dRu3 · 2026-03-12

**Soundness:** 3
**Presentation:** 4
**Significance:** 3
**Originality:** 3
**Overall Recommendation:** 5
**Confidence:** 4

**Summary:**

State-space models (SSMs) are computationally efficient alternatives to Transformers, but they underperform on tasks requiring in-context retrieval. Prior work identified that this gap stems from a small set of specialized "Gather-and-Aggregate" (G&A) attention heads that SSMs cannot replicate.

This paper introduces retrieval-aware distillation, a method for converting a pretrained Transformer into a hybrid Transformer-SSM model by surgically preserving only these retrieval-critical heads. The process has two steps: 1. each attention head is scored by measuring how much accuracy drops when it is ablated on a synthetic key-value retrieval task; 2. the top-scoring heads are kept, while all others are replaced with SSM-based recurrent heads. The resulting model is distilled from the original Transformer teacher.

The paper experimented over Llama-3.2-1B and Qwen2.5-1.5B over various knowledge-focused and retrieval-heavy datasets.

**Compliance With Llm Reviewing Policy:**

Affirmed.

**Final Justification:**

I still have remaining concerns and questions about how well this approach generalizes to different selection criteria and how the method performs on realistic long-context benchmarks such as LongBench in the rebuttal (W1) as mentioned in rebuttal acknowledgement. Thus, I will keep my score.

**Key Questions For Authors:**

Q1: Does the retrieval-aware head selection strategy generalize to other distillation frameworks beyond MOHAWK?

Q2. Have the authors compared the performance with different hybrid architectures without distillation, such as Hymba [1] or Samba [2], which combine attention and SSM layers?

[1] Hymba: A Hybrid-head Architecture for Small Language Models

[2] Samba: Simple Hybrid State Space Models for Efficient Unlimited Context Language Modeling

**Limitations:**

yes

**Strengths And Weaknesses:**

## Strength

S1. Clear motivation and simple method: The motivation is clear and the proposed method is simple and intuitive. The paper is nicely written, making it easy to follow and reproduce.

S2. Strong empirical results: The empirical results are strong, recovering over 95% of teacher performance with only 2% of attention heads while reducing inference memory by up to 6×.

S3. Thorough ablations: The paper provides thorough ablations over the number of retained heads, state dimension size, and head placement strategies, giving good insight into each design choice.


## Weakness

W1. Evaluation on more realistic retrieval benchmarks: While the paper evaluates on several retrieval-heavy benchmarks, most are synthetic or relatively simple. Evaluation on more realistic long-context benchmarks such as LongBench [1] would better validate the method's practical effectiveness and generalizability.

W2. Sensitivity to the head selection threshold: The paper sweeps over the number of retained heads k, but does not analyze sensitivity to the scoring threshold used to rank heads. It is unclear how robust the method is to this choice, and whether a different threshold or selection criterion would lead to meaningfully different architectures or performance outcomes.

[1] LongBench v2: Towards Deeper Understanding and Reasoning on Realistic Long-context Multitasks

---

> ### Author Rebuttal · Authors · 2026-03-31
>
> We appreciate the positive assessment and are glad the motivation, empirical results, and ablations all landed clearly. Below, we address each question in turn.
>
> > Does the retrieval-aware head selection strategy generalize to other distillation frameworks beyond MOHAWK?
>
> Yes. The retrieval-aware head selection is not specific to MOHAWK; It is a general criterion for identifying which attention heads to preserve when constructing a Transformer–SSM hybrid. We instantiate it within MOHAWK because it provides a strong distillation framework, but the selection procedure itself is framework-agnostic. We will revise the paper to delineate more clearly between our head-selection mechanism and the particular distillation pipeline used in our experiments.
>
> > Have the authors compared the performance with different hybrid architectures without distillation, such as Hymba or Samba, which combine attention and SSM layers?
>
> Our current comparisons focus on hybrids that are directly comparable to our setting, namely methods that differ in how attention is retained from a given teacher while using the same distillation pipeline. This isolates the architectural effect of retrieval-aware preservation. Comparing against independently trained hybrids such as Hymba or Samba would introduce additional confounders, including differences in pretraining data, model scale, training recipe, and optimization budget, making it difficult to attribute any performance gap to the hybridization strategy itself. We will clarify this framing in the paper.
>
> > The paper sweeps over the number of retained heads k, but does not analyze sensitivity to the scoring threshold used to rank heads. It is unclear how robust the method is to this choice, and whether a different threshold or selection criterion would lead to meaningfully different architectures or performance outcomes.
>
> We agree that the threshold sensitivity should be clarified. Since we select the top-k heads by rank, any threshold value that falls between the k-th and (k+1)-th ranked scores yields the same architecture. Sweeping over k therefore fully characterizes all distinct threshold choices, and Table 1 directly reports performance at each k. We will make this point explicit in the paper.
>
> We also agree that the selection criterion is not unique. Different scoring rules could lead to different head subsets, hybrid architectures, and performance outcomes, and we made this limitation explicit in the conclusion. Our current criterion is motivated by prior work [1] suggesting that retrieval is a central factor behind the Transformer–SSM gap, so we focus on retrieval heads. Exploring alternative scoring criteria is a salient direction for future work.
>
>
> ### References:
> - [1] Bick et al., arXiv:2504.18574

---

> > ### Author Rebuttal · Reviewer_dRu3 · 2026-04-03
> >
> > Thank you for the response. However, questions remain about how well this approach generalizes to different selection criteria. Additionally, while the authors conducted retrieval-heavy benchmarks, they do not discuss how the method performs on realistic long-context benchmarks such as LongBench in the rebuttal (W1). Therefore, I will keep my score.

---

### Decision · Program_Chairs · 2026-04-30

**Decision:**

Accept (regular)

**Comment:**

All reviewers are positive about the work. Therefore, I recommend an acceptance.